# Differential mitochondrial protein interaction profile between human translocator protein and its *A147T* polymorphism variant

**Prita R. Asih**[1¤], **Anne Poljak**[2], **Michael Kassiou**[3], **Yazi D. Ke**[1], **Lars M. Ittner**[1]*

**1** Dementia Research Centre, Faculty of Health and Medical Sciences, Macquarie University, Sydney, NSW, Australia, **2** Mark Wainwright Analytical Centre, University of New South Wales, Sydney, Australia, **3** School of Chemistry, Faculty of Science, University of Sydney, Sydney, NSW, Australia

¤ Current address: Flinders Health and Medical Research Institute, College of Medicine & Public Health, Flinders University, Adelaide, SA, Australia
* lars.ittner@mq.edu.au

## Abstract

The translocator protein (TSPO) has been implicated in mitochondrial transmembrane cholesterol transport, brain inflammation, and other mitochondrial functions. It is upregulated in glial cells during neuroinflammation in Alzheimer's disease. High affinity TSPO imaging radioligands are utilized to visualize neuroinflammation. However, this is hampered by the common A147T polymorphism which compromises ligand binding. Furthermore, this polymorphism has been linked to increased risk of neuropsychiatric disorders, and possibly reduces TSPO protein stability. Here, we used immunoprecipitation coupled to mass-spectrometry (IP-MS) to establish a mitochondrial protein binding profile of wild-type (WT) TSPO and the A147T polymorphism variant. Using mitochondria from human glial cells expressing either WT or A147T TSPO, we identified 30 WT TSPO binding partners, yet only 23 for A147T TSPO. Confirming that A147T polymorphism of the TSPO might confer loss of function, we found that one of the identified interactors of WT TSPO, 14-3-3 theta (YWHAQ), a protein involved in regulating mitochondrial membrane proteins, interacts much less with A147T TSPO. Our data presents a network of mitochondrial interactions of TSPO and its A147T polymorphism variant in human glial cells and indicate functional relevance of A147T in mitochondrial protein networks.

## Introduction

The 18kDa translocator protein (TSPO) [1] is a five transmembrane domain protein that resides in the outer mitochondrial membrane (OMM), at contact sites between the outer and inner mitochondrial membrane [2]. TSPO binds cholesterol with high affinity and possibly facilitates its transport across the OMM [3]. However, its physiological role *in vivo* remains elusive. It has previously been reported that *TSPO* null mice present without overt phenotypes [4].

TSPO is highly conserved across species [5] and is ubiquitously expressed [6], with higher levels in cells and organs dedicated to steroid synthesis, such as the adrenal gland cortex [7].

**Funding:** This works was supported by funding from the National Health and Medical Research Council (Grant numbers 1136241, 1132524, 2001572) and the Australian Research Council (DP210101957). The funders had no role in study design, data collection and analysis, decision to publish, or preparation of the manuscript.

**Competing interests:** The authors have declared that no competing interests exist.

Interestingly, several brain conditions, including Alzheimer's disease (AD), Parkinson's disease (PD) and traumatic brain injury (TBI), show a significant increase in expression levels of TSPO, reflecting microglial activation that is likely to result in the production of protective neurosteroids, as a homeostatic response [8]. This has been utilized diagnostically with the development of a large number of TSPO ligands used for imaging with positron emission tomography (PET) [9, 10]. TSPO has furthermore been suggested as a drug target for neurodegenerative conditions [11] and may also be of therapeutic interest in cancer, as TSPO ligands reduce proliferation of cancer cells [12].

A single nucleotide polymorphism (*rs6971*) in the *TSPO* gene, which resulted in the substitution of alanine by threonine at position 147 (*A147T*) in the helical turn before the cholesterol binding motif, has hampered the use of TSPO radio-labelled ligands as an active biomarker for inflammation by *in vivo* visualization (PET imaging) [13]. This one point mutation is observed predominantly in Caucasian populations, in which approximately 10% are homozygous and up to 30% are heterozygous [14]. The *A147T* polymorphism not only decreases the binding affinities of many TSPO ligands but has also been shown to affect pregnenolone production and was found to be associated with neurological diseases like bipolar disorder [15, 16]. However, these findings were indirectly demonstrated through the actions of its ligands. To obtain insight into the functional relevance of this *A147T* point mutation, we asked how this mutation might impact the protein interactome of TSPO within the mitochondrial complex. Therefore, we transfected human glioma cell U87MG with full length human $TSPO^{WT}$ or its *A147T* variant (both with C-terminal V5 tags) and isolated mitochondria from these cells. These mitochondria enriched fraction was subsequently immunoprecipitated for the tagged proteins and further subjected to mass spectrometry (IP-MS). Selected interaction partners were further validated by co-immunoprecipitation.

## Materials and methods

### Mitochondria isolation

Mitochondria were isolated from U87MG cells based on a previously described protocol [17]. Briefly, cell pellets were resuspended in 1 mL aliquots of ice-cold RSB hypo buffer (10 mM NaCl, 1.5 mM $MgCl_2$, 10 mM Tris-HCl (pH 7.5) containing complete EDTA-free protease inhibitor) and transferred to a 2-mL Dounce homogenizer. Cells were allowed to swell for 5–10 min. The progress of the swelling was checked using a phase-contrast microscope. The swollen cells were lysed with several strokes of the B pestle, pressed straight down the tube with a firm, steady pressure. Immediately 500µL of ice-cold 2.5× MS homogenization buffer (525 mM mannitol, 175 mM sucrose, 12.5 mM Tris-HCl (pH 7.5), and 2.5 mM EDTA (pH 7.5) with protease inhibitor) was added to each sample to obtain a final concentration of 1× MS homogenization buffer. Subsequently, the top of the homogenizer was covered with Parafilm and mixed by inverting twice. A portion of the homogenate was retained to perform enzyme marker assays. Then the homogenate was transferred to a centrifuge tube for density gradient centrifugation. The final volume was brought to 2 mL with ice-cold 1× MS homogenization buffer (210 mM mannitol, 70 mM sucrose, 5 mM Tris-HCl (pH 7.5), and 1 mM EDTA (pH 7.5) with protease inhibitor). Afterwards, the homogenate was centrifuged at 1300×**g** for 5 min to remove nuclei, unbroken cells, and large membrane fragments. The supernatant was poured into a clean centrifuge tube and centrifuged again at 1300×g for 5 min. This centrifugation step was repeated two more times. The supernatant from the third centrifugation was transferred to a clean centrifuge tube and centrifuged at 17,000×g for 15 min to obtain the mitochondria (pellet). The isolated mitochondria were washed in ice-cold 1× MS buffer and

centrifuged at 17,000xg. The supernatant was discarded, and the resulting pellet (mitochondria) was resuspended in NP40 buffer containing protease inhibitor.

## Immunoprecipitation

Immunoprecipitation (IP) of transfected cells was carried out as previously described [18]. Following harvesting the transfected cells and isolating the mitochondria, the sample(s) were pre-incubated with protein G-coupled magnetic beads (Invitrogen) for 1 h. Samples were then incubated with antibodies against V5 overnight at 4˚C. The following day, the samples were incubated with protein G-coupled magnetic beads (1 h at 4˚C on a rotating platform) and then washed 4 times with IP buffer (50mM HEPES (pH 7.5), 140mM sodium chloride, 0.2% (v/v) NP40 with protease inhibitors). The precipitate was recovered from the beads by boiling in hot (95˚C) sample buffer (1M Tris-HCl (pH 8.0), 20% (v/v) β-mercaptoethanol, 40% (v/v) glycerol, 9.2% (w/v) SDS and 0.2% (w/v) bromophenol blue) and proteins were separated by SDS-PAGE, Western blot transferred to PVDF membrane, and probed for Myc (1:5000) and V5 (1:5000). The second approach of IP of transfected cells was carried out by immunoprecipitation with antibodies against respective interacting proteins. The subsequent procedure was similar to that outlined above, and immunoblotting was carried out with V5 (1:5000).

## Immunoblotting

For SDS PAGE and Western blot analysis of TSPO transfected cells, 10μg protein was loaded onto SDS PAGE gels as the input and the entire immunoprecipitate for each sample was loaded in successive lanes. Proteins were subsequently electrophoretically transferred onto nitrocellulose membranes (GE Healthcare) in a Trans-Blot SD Semidry Transfer Cell (BioRad) at 25V for 45 min. Membranes were subsequently blocked in 5% (w/v) BSA in Tris-buffered saline containing 0.1% (v/v) Tween20 (TBS-T) for 1 h at ambient temperature, followed by incubation with the primary antibody diluted in 5% (w/v) BSA/TBS-T (overnight at 4˚C on a shaking platform). For IP experiments using TSPO-transfected cells, the primary antibodies used were to V5 (1:5000), myc (1:5000), VDAC1 (1:1000), Hsp90AA1 (1:2000), YWHAE (1:1000), YWHAZ (1:1000), or YWHAG (1:1000). The following day the primary antibody was removed, the membranes washed three times with TBS-T and incubated with alkaline phosphatase-coupled secondary antibodies (1:10,000, Sigma) in 1% (w/v) BSA/TBS-T for 45 min at room temperature. Protein bands were visualized with Immobilon Chemiluminescent Alkaline Phosphatase substrate (Millipore) and detected in a VersaDoc Model 4000 CCD camera system (BioRad). To determine equal loading and for normalization, membranes were stripped by washing in distilled water for 5 min, followed by 0.2M sodium hydroxide for 10 min and then distilled water for 5 min after which they were reprobed with antibodies against GAPDH.

## Expression plasmids

To obtain expression constructs for transient transfection of U87MG cells, the coding sequence encoding human TSPO (*hTSPO*) was amplified from cDNA previously generated by reverse transcription from total HEK293 mRNA isolated using Trizol (Life Technologies) and cloned into either a pENTR/SD/TOPO vector (Invitrogen) or pGEM-T-Easy vector (Promega). For cloning into pGEM-T-Easy vector, *hTSPO* was amplified by PCR using PfuTurbo High-Fidelity DNA Polymerase (Agilent Technologies). A polyA tail was added to this blunt-end PCR product and ligated into the pGEM-T-Easy vector overnight at 4˚C. Ligated vectors are transformed into *One Shot Top10 E. coli* for pENTR/SD/TOPO, *DH5αE. Coli* for pGEM--T-Easy vectors and plated out onto plates containing 50 μg/mL kanamycin (pENTR/SD/

TOPO) and 100 μg/mL ampicillin (pGEM-T-Easy). Positive clones were identified via restriction digest and confirmed via DNA sequencing.

## Cloning and site-directed mutagenesis

Site-directed mutagenesis was carried out to introduce the *A147T* point mutation into the *hTSPO* cDNA according to the Agilent Technologies' Quikchange XL Site-directed Mutagenesis Kit instructions. Briefly, mutagenic oligonucleotide primers incorporating the *A147T* point mutation were used to introduce the mutation into *hTSPO* in pGEM-T-Easy. Twenty-five cycles of PCR were carried out in a final volume of 50μl, containing 1ng of *hTSPO* in pGEM-T-Easy as template, 10pmol of forward and reverse primers, and 2.5 units of *PfuTurbo*. The annealing temperature was set to 58˚C. The PCR products generated were digested with 10 units of *Dpn* I restriction enzyme at 37˚C for 3 h to digest the parental DNA template. Digested PCR products were transformed into *XL1 blue* competent cells and plated onto agar plates containing 100 μg/mL Ampicillin. Positive clones were confirmed via DNA sequencing analysis.

## Transformation of plasmid DNA

Chemically competent *E. coli* cells (100 μl) were thawed on ice and transferred into a chilled 14 mL polypropylene round-bottom tube (BD Biosciences). Then 50ng of the desired plasmid DNA was added to the *E. coli* and the *E. coli*/DNA mixture was incubated for 20–30 min on ice. Subsequently, this *E. coli*/DNA mix was heat-shocked for 40 seconds in a 42˚C water bath and cooled on ice for 2 min. S.O.C medium (250 μl, Life Technologies) was added and tubes shaken at 200 rpm for 1 h at 37˚C. The transformed *E. coli* were plated onto Lysogeny Broth (LB) agar plates with the appropriate antibiotics and incubated overnight at 37˚C. For subsequent analysis of clones obtained, single colonies were picked and inoculated in LB medium containing ampicyline, incubated overnight at 37˚C and plasmid DNAs were purified using the Promega Wizard mini-prep kit.

## Transient transfection

The polyethylenimine (PEI) method of transfection was used to deliver plasmid DNA into U87MG cells in a 10 cm dish as previously described [19] with minor modifications. Briefly, cells were plated at the density of 0.5 x 10$^6$ per 10 cm dish and allowed to become adherent. The next day, when cells reached 60–80% confluency, the media was changed 30 min before transfection. Then, the PEI-DNA mixture was prepared using 3:1 ratio of PEI to DNA (w/w). Briefly, 30 μg of PEI (1 g PEI dissolved in 1000 mL distilled water pH7) was diluted into a total volume of 750 μl of 0.9% NaCl. Then, 10 μg of DNA was diluted into a total volume of 750 μl of 0.9% NaCl. Subsequently, the diluted DNA was added to the diluted PEI and incubated at ambient temperature for 15 min. The PEI-DNA mixture was then added dropwise to the 10 cm dish and left for 48 h, after which the cells were ready to be harvested in NP-40 lysis buffer with protease inhibitor.

## Mass-spectrometry and data analysis

Briefly, IP samples were desalted, and buffer exchanged using 3-kDa Amicon filter units (EMD Millipore) using 50 mM sodium bicarbonate. Protein sample (400 μg) was reduced by adding 2 μl of Tris(2-carboxyethyl)phosphine (Sigma) and incubating samples at 60˚C for 1 h. Samples were then alkylated by adding 1 μl of iodoacetamide (37 mg/mL) to block cysteine side chains (10 min, ambient temperature). Protein samples were digested using 4 μg of trypsin

(sequencing grade, low autolysis trypsin; Promega), at 37˚C for 16 h. Digested samples were briefly spun in a microcentrifuge and pH checked and, if necessary, adjusted to pH 9–10 with a few μL of sodium carbonate (500 mM $Na_2CO_3$). Samples were analyzed in biological triplicates (6 μg injected per run) by LC-MS/MS (HCD on the QExactive Plus: Thermo Electron, Bremen, Germany), followed by Mascot searching to identify peptides and proteins. Adaptations of previously published protocols were used [20, 21].

Only proteins with ≥2 annotated peptides in Mascot were used for further analysis. The Mascot Score is a statistical score for how well the experimental data match the database sequence [22], and Mascot parameters included using 15-ppm parent mass tolerance, 0.5 Da fragment mass tolerance, and Protein False Discovery Rate equal or lower than 1%. Searches were conducted against several databases such as Uniprot, Swissprot, and NCBI. Mascot results are statistically interpreted using a Java-based re-implementation of the Peptide Prophet Algorithm [23], which converts search engine scores into probabilities of peptide identification. Data compilation and quantification across 3 IP biological replicates for each of the experimental groups were performed using Scaffold 2.4.0 and a 95% protein probability threshold was applied. Scaffold computes protein grouping at two levels: first to determine the identified protein groups within a single MS/MS sample, and again across all MS/MS samples in the experiment to determine an experiment wide protein identification list [24]. Scaffold uses a reimplementation of the Peptide Prophet algorithm to combine peptides into protein identifications using a statistical model that considers a distribution of the number of peptides assigned to each protein. Quantitative analysis from ScaffoldQ was utilized by averaging over the 3 biological replicates and comparisons applying a *p* value (Student's *t* test) criterion of $p < 0.05$ to determine the statistically significant differentially bound proteins. Protein ontology was analyzed using DAVID gene ontology database (v6.8) using gene ontology annotations for the entire human genome as the background list. Interactomes and network properties were visually represented using CytoScape (v3.7.1). Existing evidence for protein–protein interactions of TSPO was derived from BioGRID (v3.1.180) and STRING (v10.5) databases and through literature searching.

## Results

### Identification of the hTSPO^WT and hTSPO^A147T interacting proteins in human glioma cells

Interactions of TSPO with other proteins have been characterized by conventional methods, such as in-gel digestion followed by mass-spectrometry (MS) analysis [25] or pull-down methods [26] which are limited by antibody affinity, lysis, and solubility conditions. Moreover, identification of interaction partners of the cholesterol-binding domain of TSPO (the CRAC domain) have been identified by coimmunoprecipitation [27]. However, identification of TSPO interaction partners in the context of the frequent *A147T* polymorphism have not been studied. To gain insight into the effects of the *A147T* polymorphism on mitochondrial TSPO interactions, we employed immunoprecipitation coupled to mass-spectrometry (IP-MS). The use of IP-MS is considered the gold standard in regards to sensitivity and specificity for the identification of protein interactions (complexes) [28]. We tagged full-length human *TSPO^WT* and *TSPO^A147T* with a C-terminal V5 tag for expression in U87MG cells, a human astro-/ microglioma cell line of CNS origin [29]. V5-tagged TSPO protein levels were equally transfected for both *TSPO^WT* and *TSPO^A147T* (**Fig 1A**). Isolated mitochondria from these cells showed enrichment of the mitochondrial protein voltage-dependent anion channel 1 (VDAC1) and absence of the nuclear protein heterogenous nuclear ribonucleoproteins (hnRNP), indicating successful purification of mitochondria (**Fig 1B–1D** and Supporting

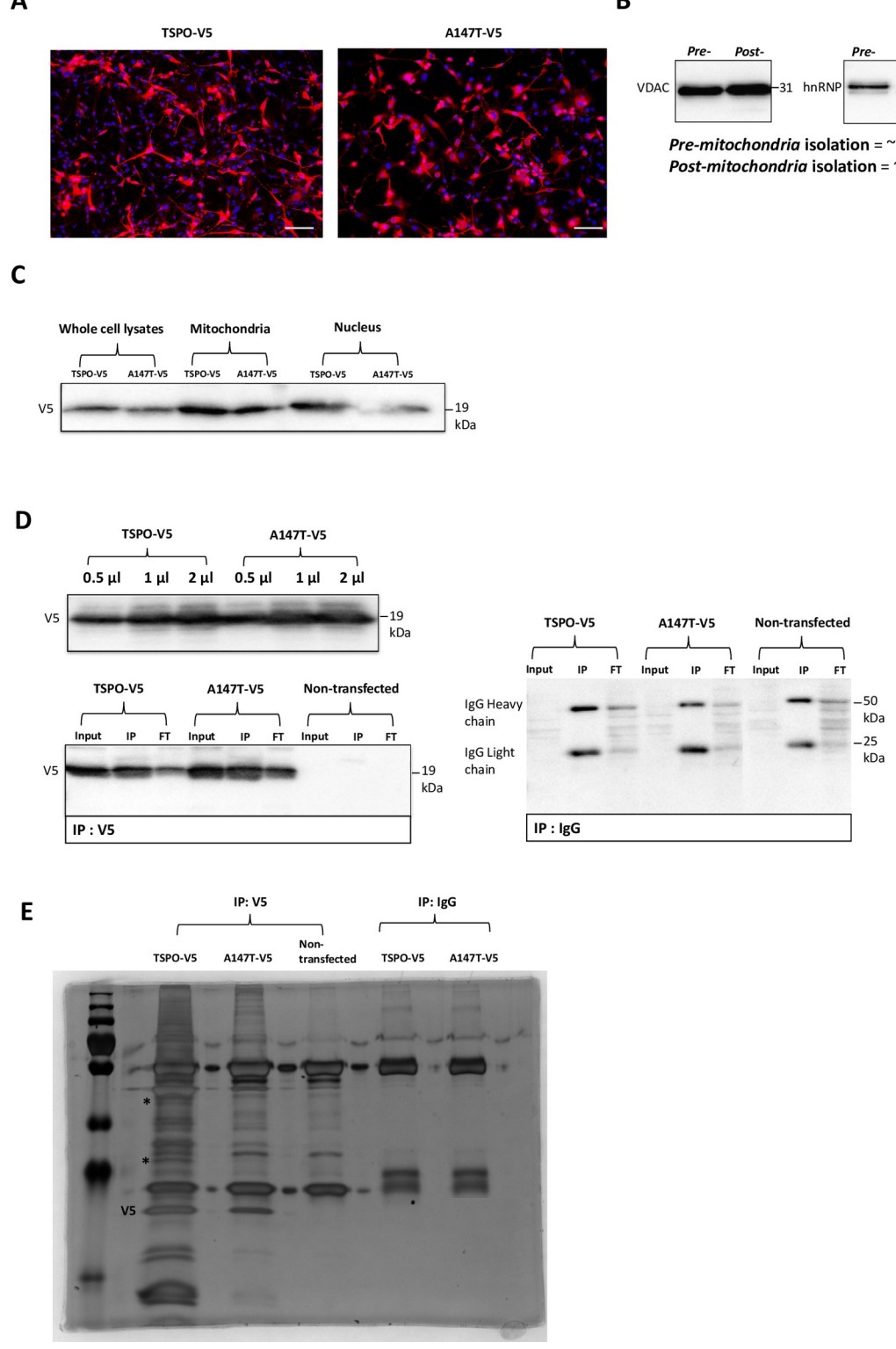

**Fig 1. Optimization of sample preparation for the identification of binding partners between human TSPO and its mutant *A147T* by IP-MS.** A) Transfection efficiency of V5-tagged TSPO$^{WT}$ and TSPO$^{A147T}$ in U87MG cells (42.6% ± 5.93), indicating equal protein expression levels; *red*, anti-V5 immunofluorescence and *blue*, DAPI indicating cell nuclei. n = 3. B) Mitochondrial enrichment of U87MG cells as assessed by Western blot for VDAC as a mitochondrial marker protein and hnRNP as a nuclear marker protein, indicating that purified mitochondria were isolated, n = 2. C) Representative immunoblot of mitochondrial lysates from transfected U87MG cells with V5-tagged

TSPO$^{WT}$ and TSPO$^{A147T}$. This confirms equal expression levels of V5 in both TSPO$^{WT}$ and TSPO$^{A147T}$ extracts. D) Immunoprecipitation of TSPO$^{WT}$ and TSPO$^{A147T}$ from transfected U87MG cells with increasing amount of mouse V5 antibody, with 2μg per 400 μg of total protein chosen for subsequent experiments. Non-specific mouse IgG antibody was used as a negative control. Immunoblot of input (10% of the total lysates) confirms comparable expression levels of V5 tag between TSPO$^{WT}$ and TSPO$^{A147T}$. Immunoblot of unbound V5 tag shows the amount of non-precipitated TSPO. E) Immunoprecipitation of TSPO$^{WT}$ and TSPO$^{A147T}$ from transfected U87MG cells. Precipitated proteins were visualized by silver staining of SDS-PAGE.

information for uncropped blots for the referenced western blots figures throughout this study with areas displayed in the main figures highlighted as red broken boxes). Furthermore, equal expression levels of V5-tagged TSPO$^{WT}$ and TSPO$^{A147T}$ was confirmed in isolated mitochondria by the immunoblotting (**Fig 1C**). Optimization of the V5 antibody for IP showed that 2 μl of antibody per 400 μg total protein was the optimum condition to pull down both V5-tagged TSPO$^{WT}$ and TSPO$^{A147T}$ (**Fig 1D**). To assess the specificity of V5 antibody and to control for non-specific binding, both isolated mitochondria of TSPO$^{WT}$ and TSPO$^{A147T}$ expressing cells as well as non-transfected cells were immunoprecipitated using IgG antibody that served as negative control (**Fig 1D**). Immunoblot of input (10% of the total lysates) further confirmed comparable expression levels between TSPO$^{WT}$ and TSPO$^{A147T}$ whereas immunoblot of unbound V5-tagged proteins (FT = flow through) showed a relatively small amount of non-precipitated TSPO (**Fig 1D**). Next, one third of the IP samples were run by SDS PAGE, silver stained, and the result indicated that TSPO$^{A147T}$ had fewer bands and/or weaker intensity compared to TSPO$^{WT}$ samples (**Fig 1E**). Overall, these results show that V5-tagged TSPO expression in TSPO$^{WT}$ and TSPO$^{A147T}$ yields similar expression levels in isolated mitochondria of U87MG cells, which supported efficient and comparable immunoprecipitation for subsequent MS.

To identify the interaction partners of V5-tagged TSPO$^{WT}$ and TSPO$^{A147T}$, we performed MS of enriched mitochondrial fractions from U87MG cells at sub-confluency and transfected with constructs expressing either V5-tagged *TSPO$^{WT}$*, V5-tagged *TSPO$^{A147T}$*, or non-transfected cells. A schematic outline of the IP-MS analysis procedure and results is shown in **Fig 2A**. For protein identification, immunoprecipitated proteins were trypsin digested in-solution and the resulting peptides were identified by liquid chromatography tandem mass spectrometry (LC-MS/MS). The mass spectrometric data contain a list of the accurate peptide masses and peptide fragment masses. These were searched against sequence databases containing known protein amino acid sequences using the Mascot software package. Based on Mascot LC-MS/MS search engine (Swissprot Database, human taxonomy), 647 unique proteins were identified, with a protein False Discovery Rate ≤1% and were identified in TSPO$^{WT}$, TSPO$^{A147T}$, and non-transfected (WT) mitochondria that were immunoprecipitated by either V5 antibodies or control IgG (**Fig 2A**, **S1 Table**). Further manual analysis using Scaffold software was carried out to exclude peptide sequences that were present in both the wild-type TSPO samples as well as the IgG controls, yielding a final list of 307 proteins expressed specifically in TSPO$^{WT}$, TSPO$^{A147T}$, samples, but not expressed (or minimally) in the controls. Of these, 44 proteins were known mitochondrial proteins, 51 proteins were of cytoplasmic localization, and the rest were nuclear GO annotated proteins. Subsequently, further manual stringent measures were taken to remove peptide sequences that were only present in less than 2 out of 3 IP TSPO$^{WT}$ and TSPO$^{A147T}$ replicates. This further limited the number of candidate proteins to 56 for TSPO$^{WT}$ and 47 for TSPO$^{A147T}$. A significant difference ($p<0.01$-$p<0.1$) in total spectral counts between three biological replicates of each of TSPO$^{WT}$ and TSPO$^{A147T}$ was a further selection criterion applied to identify TSPO binding partners. Twenty-three candidate proteins interacted with both TSPO$^{WT}$ and TSPO$^{A147T}$, and 7 candidate proteins interacted only with TSPO$^{WT}$ (**Fig 2A**). A functional protein association network analysis of the identified TSPO interaction partners was generated using

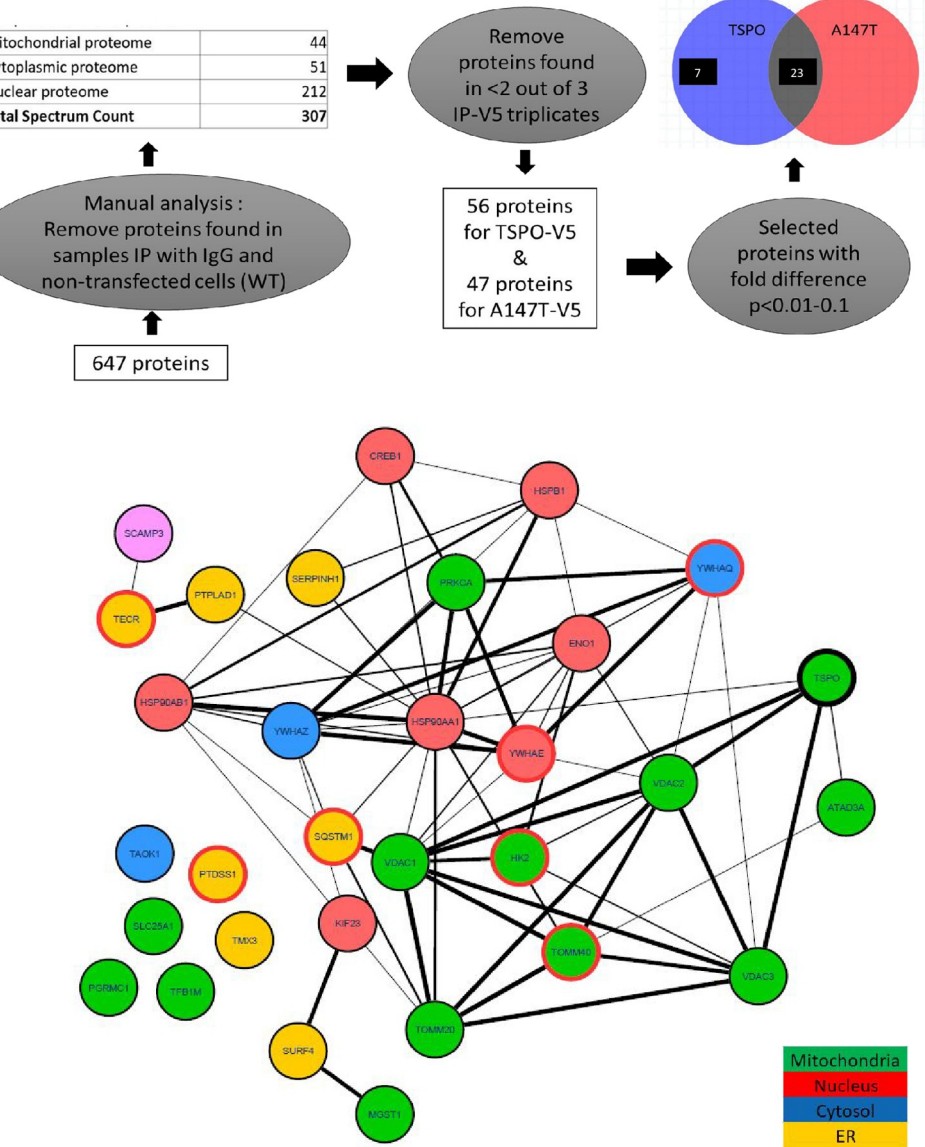

**Fig 2. Identification of binding partners of hTSPO$^{WT}$ and hTSPO$^{A147T}$ by label free semiquantitative proteome analysis.** A) Diagram of the immunoprecipitation-mass spectrometry approach to identify the TSPO$^{WT}$ and TSPO$^{A147T}$ interactomes. We identified 56 proteins for V5-tagged human TSPO$^{WT}$ and 47 proteins for human TSPO$^{A147T}$, of which two thirds comprised of mitochondrial proteins and the remainder were cytoplasmic and nuclear proteins. Of these, 23 selected candidates interacted with both TSPO$^{WT}$ and TSPO$^{A147T}$ and 7 candidates interacted only with TSPO$^{WT}$, after further selection criteria with protein fold difference of $p<0.01$–0.1 B) A functional protein association network generated using STRING (v10.5) and Cytoscape (v6.3.0) bioinformatics software. Network edges represent protein-protein associations and represent confidence, with the strength of the association represented by line thickness. The circles (nodes) depict specific proteins, and the color represents sub-cellular localization of each protein, as shown in the legend. The 7 protein candidates which interact solely with TSPO$^{WT}$ are indicated on nodes with red outlines. Nodes with no network edges represent proteins that have never previously been reported as TSPO interaction partners.

STRING and Cytoscape bioinformatics software and are shown in **Fig 2B**. All of these candidate proteins were in the top 50% of the highest interaction probability and abundance (spectral counts). Of note are known interacting partners of TSPO, including VDAC1, VDAC2, VDAC3,

**Table 1. Functional annotation cluster analysis of TSPO binding partners extracted from from V5-tagged TSPO$^{WT}$ and TSPOA$^{147T}$-transfected samples and detected by MS.** Top 3 DAVID Bioinformatics Resources annotation clusters (n = 31 proteins from three biological replicates). Fold change refers to enrichment in comparison to the whole human proteome (background list).

| Biological process | Gene count | p-Value | Genes | Fold change | FDR |
|---|---|---|---|---|---|
| Annotation cluster 1. Enrichment score 4.41 | | | | | |
| Mitochondrial outer membrane | 8 | 3.7 x $10^{-9}$ | HK2, MGST1, TOMM20, TOMM40, **TSPO**, VDAC1, VDAC2, VDAC3 | 32 | 4 x $10^{-6}$ |
| Mitochondria | 12 | 2.5 x $10^{-7}$ | ATAD3A, HK2, MGST1, PRKCA, SLC25A1, TFB1M, TOMM20, TOMM40, **TSPO**, VDAC1, VDAC2, VDAC3 | 7.1 | 2.7 x $10^{-4}$ |
| Anion transport | 4 | 1.7 x $10^{-6}$ | **TSPO**, VDAC1, VDAC2, VDAC3 | 160 | 2.2 x $10^{-3}$ |
| HLTV-I infection | 5 | 6.6 x $10^{-3}$ | CREB1, **TSPO**, VDAC1, VDAC2, VDAC3 | 6.2 | 7.2 |
| Transport | 8 | 2.1 x $10^{-2}$ | SCAMP3, SLC25A1, TOMM20, TOMM40, **TSPO**, VDAC1, VDAC2, VDAC3 | 2.7 | 2.1 |
| Annotation cluster 2. Enrichment score 3.3 | | | | | |
| Ion channel binding | 6 | 1.5 x $10^{-6}$ | HSP90AA1, HSP90AB1, **TSPO**, YWHAE, YWHAQ, VDAC1 | 29 | 1.8 x $10^{-3}$ |
| Annotation cluster 3. Enrichment score 3.14 | | | | | |
| Protein targeting | 4 | 4.1 x $10^{-5}$ | TOMM20, YWHAE, YWHAQ, YWHAZ | 57 | 5.5 x $10^{-2}$ |

Abbreviations: FDR, False Discovery Rate; MS, mass spectrometry; HLTV-I, Human T-cell leukemia virus type 1.

and ATAD3A that were also identified in our IP-MS experiments, increasing confidence in our approach (**Fig 2B**).

Analysis of the TSPO$^{WT}$ and TSPO$^{A147T}$ interactomes by protein domain ontology identified an enriched set of mitochondrial outer membrane proteins (**Table 1**. DAVID INTERPRO domain enrichment cluster 1, p = 3.7 x 10–9) and ion channel binding proteins (**Table 1**. DAVID INTERPRO domain enrichment cluster 2, p = 1.5 x 10–6). Mitochondrial outer membrane proteins constituting cluster 1 were VDAC1, VDAC2, VDAC3, HK2, MGST1, TOMM20, and TOMM40 (**Table 1**), whereas ion channel binding proteins constituting cluster 2 were HSP90AA1, HSP90AB1, YWHAE, YWHAQ, and VDAC1 (**Table 1**).

Our IP-MS results identify interactions of TSPO$^{WT}$ and TSPO$^{A147T}$ in cultured glial cells with proteins of heterogeneous functions, including protein transmembrane transport into intracellular organelle, ion channel binding and regulation of protein targeting, possibly explaining the multiple functions of TSPO in cells. Importantly, analysis of the TSPO$^{A147T}$ interactome revealed the loss of interactions with 7 proteins as compared to TSPO$^{WT}$, namely mitochondrial import receptor subunit TOM40 homolog (TOMM40), 14-3-3 protein subunit epsilon (YWHAE), 14-3-3 protein subunit tau/theta (YWHAQ), hexokinase 2 (HK2), sequestosome 1 (SQSTM1), phosphatidylserine synthase 1 (PTDSS1), and trans-2,3 enoyl-CoA reductase (TECR) (**Fig 2B**). Taken together, the *A147T* point mutation of TSPO might result in a loss of interacting partners important for mitochondrial and cell function.

## Validation of selected human TSPO protein-protein interactions

Specific interactions of several identified candidate binding partners of human TSPO$^{WT}$ and TSPO$^{A147T}$ were confirmed by co-immunoprecipitation (co-IP) from transiently transfected

cells followed by immunoblotting. The chosen candidate binding partners were voltage-dependent anion-selective channel protein 1 (VDAC1), heat shock protein HSP90-alpha (HSP90AA1), 14-3-3 protein subunit theta (YWHAQ) and sequestosome-1 (SQSTM1). Two approaches of co-IP were employed. The first approach involved pull-down of cells overexpressing either V5-tagged TSPO$^{WT}$ or TSPO$^{A147T}$, with antibodies against their respective interacting proteins and then immunoblotted for TSPO with V5 antibodies. The candidate proteins validated with this approach were VDAC1 and HSP90AA1. Cells overexpressing V5-tagged pcDNA was used as a negative control. As expected from our MS results, there were no differences in VDAC1 (**Fig 3A**) and HSP90AA1 interaction (**Fig 3B**) with either TSPO$^{WT}$ or TSPO$^{A147T}$, respectively.

The second approach involved co-IP with V5 antibody, of cells overexpressing either V5-tagged TSPO$^{WT}$, TSPO$^{A147T}$ or pcDNA together with their respective interacting proteins tagged with Myc. The candidate proteins validated with this approach were 14-3-3 protein subunit theta (YWHAQ) and sequestosome-1 (SQSTM1). TSPO$^{A147T}$ had significantly lower interaction of YWHAQ compared to TSPO$^{WT}$, indicating weaker YWHAQ interaction due to the *A147T* mutation (**Fig 3C**). Whereas there was no detectable difference in SQSTM1 interaction with TSPO between TSPO$^{WT}$ and TSPO$^{A147T}$ (**Fig 3D**). 14-3-3 protein subunit eta (YWHAH) and 14-3-3 subunit beta (YWHAB) were not on the interacting proteins list, and were therefore used as negative controls. Both YWHAH and YWHAB did not co-purified with TSPO, as expected, indicating the robustness and specificity of the mass-spectrometry list of interacting partners (**S1 Fig**).

## Discussion

Protein interactions of human full-length TSPO and its domain involving cholesterol recognition/interaction amino acid consensus (CRAC) motif have been documented in previous studies, which used blue-native polyacrylamide gel electrophoresis (BN-PAGE) coupled to MS [25] and computational modelling with *in-silico* screening [30], respectively. However, the effect of the common TSPO polymorphism *A147T* on protein interactions remains unknown. To date, only the effect of *A147T* TSPO on protein structure and stability have been reported [31, 32]. Our current study identified putative binding partners of human full-length TSPO and its *A147T* polymorphism variant using IP-MS biased towards mitochondrial proteins. Employing IP-MS of mitochondrial proteins isolated from cells overexpressing either TSPO$^{WT}$ and TSPO$^{A147T}$ and downstream rigorous peptide analysis using Scaffold software [24], we detected 30 protein-protein interactions (PPIs) of hTSPO$^{WT}$ and 23 PPIs of hTSPO$^{A147T}$ with high confidence, suggesting that the *A147T* variant of human TSPO results in an alteration of the TSPO protein interactome. The *A147T* mutation not only significantly alters the protein flexibility and stability, but also decreases the half-life of the mutant protein by about 25 percent [31]. Such structural changes would increase TSPO$^{A147T}$ turnover, and may, at least in part, explain its effect on reducing the number of protein interactions.

Among the 30 TSPO$^{WT}$ PPIs and 23 TSPO$^{A147T}$ PPIs were a number of known mitochondrial TSPO interaction partners, increasing our confidence in our proteomic data sets. These included the voltage-dependent anion-selective channel protein 1 (VDAC1) [27], voltage-dependent anion-selective channel protein 2 (VDAC2), voltage-dependent anion-selective channel protein 3 (VDAC3) [33], and ATPase family AAA domain-containing protein 3A (ATAD3A) [25]. Since we performed the IP-MS using isolated mitochondria, it is not surprising that, mitochondrial proteins were the most prevalent amongst the 30 TSPO$^{WT}$ PPIs and 23 TSPO$^{A147T}$ PPIs identified. However, we also identified a number of nuclear and cytoplasmic proteins as TSPO interaction partners. Although this may reflect impurities during

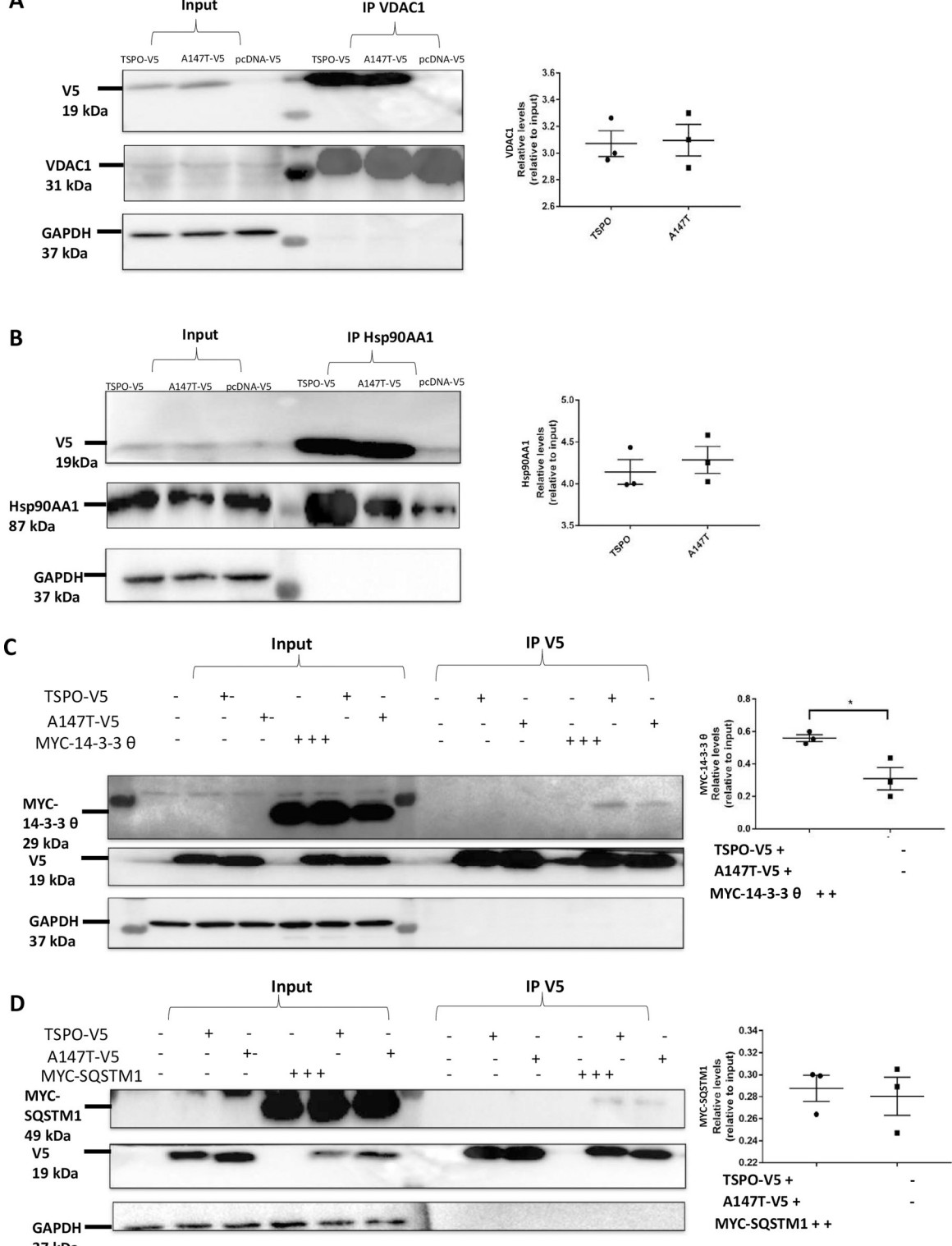

**Fig 3. Validated interactions of hTSPO^WT and hTSPO^A147T.** A) Co-immunoprecipitation (IP) of voltage-dependent anion channel 1 (VDAC1) and hTSPO^WT or hTSPO^A147T with VDAC antibody and detected by immunoblotting for V5 and VDAC1. GAPDH was used as loading control and did not co-precipitate with VDAC1. A comparable amount of V5-TSPO was co-purified with VDAC1 with each of TSPO^WT and TSPO^A147T. B) Co-IP of heat shock protein HSP90-alpha (Hsp90AA1) and hTSPO^WT or hTSPO^A147T with Hsp90AA1 antibody and detected by immunoblotting for V5 and Hsp90AA1. A comparable amount of V5-TSPO was co-purified with Hsp90AA1

with each of TSPO$^{WT}$ and TSPO$^{A147T}$. C-D) 14-3-3 θ and sequestosome 1 (SQSTM1) were expressed in U87MG cells together with V5-tagged human TSPO$^{WT}$ or TSPO$^{A147T}$. C) Co-IP of 14-3-3 θ and hTSPO$^{WT}$ or hTSPO$^{A147T}$ with V5 antibody and detected by immunoblotting for myc and V5. 14-3-3 θ was co-purified with V5 and weaker interaction was observed in TSPO$^{A147T}$ compare with TSPO$^{WT}$. D) Co-IP of SQSTM1 and hTSPO$^{WT}$ or hTSPO$^{A147T}$ with V5 antibody and detected by immunoblotting for myc and V5. Comparable amount of SQSTM1 was co-purified with TSPO$^{WT}$ and TSPO$^{A147T}$. Results from 3 independent experiments were quantified by densitometry using Image J and represented as mean ± S.D. (Student t test) * $p < 0.05$.

mitochondrial preparations, an alternative explanation is that this reflects interactions at the surface of mitochondria, where TSPO resides [34]. We found that the *A147T* polymorphism in TSPO is associated with loss of a small subset of interaction partners implicated in mitochondrial outer membrane organization, ion channel binding, and protein targeting functions. Of the confirmed interaction partners, only 14-3-3 protein subunit theta (YWHAQ) showed weaker interaction with TSPO$^{A147T}$, suggesting this point mutation modulates specifically protein interactions. Our results suggest that the *A147T* polymorphism of human TSPO partially changes its interactome, and thus its molecular function, as compared with non-mutant full-length human TSPO. Our IP-MS data suggests differential interaction of TSPO$^{WT}$ and TSPO$^{A147T}$ with outer mitochondrial membrane proteins HK2 and TOMM40 with hTSPO$^{A147T}$, as well as ion channel binding proteins and targeting proteins YWHAE and YWHAQ. Whether these normal interactions are directly with TSPO$^{WT}$ or indirectly remains to be shown. Indirect interactions of TSPO with nuclear proteins have been reported for YWHAE through VDAC1 protein [35]. Indirect interaction via VDAC1 or VDAC2 [27, 36] are unlikely, since their interaction with TSPO was not compromised by the *A147T* polymorphism. TSPO interaction with YWHAG is mediated through Ser-194 of the steroidogenic acute regulatory protein (STAR) [37]. However, YWHAG was not identified in our study, possibly due to the different cell types used by Aghazadeh and colleagues (MA10 mouse Leydig cells), that have a very different protein expression pattern compared to the human glioblastoma cells (U87MG) used in our study [38].

To our knowledge, this is the first report of YWHAQ as an interactor of TSPO$^{WT}$ and with reduced affinity to hTSPO$^{A147T}$. More than 200 proteins have been found to interact with 14-3-3 family members (there are seven isoforms in mammals: β, γ, ε, η, ζ, σ, and τ/θ), a group of proteins which are not only steroidogenesis-related proteins but also include protein kinases, enzymes, cell cycle control-related proteins, and apoptosis-related proteins [39]. In addition, 14-3-3 proteins localize to disease-specific injury sites and toxic protein aggregates in the brain, which may contribute to the regulation of disease pathogenesis [40]. As a result, 14-3-3 proteins may have either a protective role by facilitating sequestration of pathogenic proteins, or a detrimental role by promoting protein aggregate formation, neurotoxicity, and/or loss of stabilization of binding partners [41]. For example, YWHAQ and YWHAE have been found to inhibit apoptosis induced by rotenone in a cellular model of PD [42]. Based on DAVID Functional Clustering enrichment analysis, a possible role of the TSPO's interaction with YWHAQ could be the regulation of TSPO's targeting into mitochondrial membranes involved in apoptotic signaling pathways. But future studies are required to provide direct evidence. Since YWHAQ is known to inhibit apoptosis [42], and TSPO has been shown to be involved in apoptosis via its interaction with VDAC by generating ROS production which leads to activation of mitophagy [43], carriers of the *A147T* TSPO may be more prone to cellular apoptosis through its weaker interaction with YWHAQ. Understanding the role(s) of this interaction with TSPO warrants further investigation.

TSPO interaction with VDAC has been well-documented for more than two decades [36] and is consistently observed in the transduceosome that bridges the inner and outer mitochondrial membrane. This interaction is best characterized for its role in hormone-induced

steroidogenesis [44]. The direct interaction of TSPO with VDAC has been demonstrated with various methods including blue native PAGE [25], immunoprecipitation and microscopy [44] and copurification [45]. However, the mode of this interaction at the molecular level is still not well characterized, despite the advent of high-resolution crystal structures of both TSPO [46] and VDAC [47, 48]. Heat shock proteins (HSPs), including HSP90AA1, are known to play an important role in delivering proteins to the outer mitochondrial membrane (OMM) for import and facilitating translocation in an ATP dependent manner [49].

## Supporting information

**S1 Fig. Validated non-interactors of hTSPO<sup>WT</sup> and hTSPO<sup>A147T</sup>.** A-B) 14-3-3 η protein (YWHAH) and 14-3-3 β (YWHAB) were expressed in U87MG cells together with V5-tagged human TSPO<sup>WT</sup> or TSPO<sup>A147T</sup>. A) Co-IP of YWHAH and hTSPO<sup>WT</sup> or hTSPO<sup>A147T</sup> with V5 antibody and detected by immunoblotting for myc. YWHAH was not co-purified with V5. B) Co-IP of YWHAB and hTSPO<sup>WT</sup> or hTSPO<sup>A147T</sup> with V5 antibody and detected by immuno-blotting for myc. YWHAB was not co-purified with V5.
(PDF)

**S1 Table. Mascot LC-MS/MS results from immunoprecipitations of hTSPO<sup>WT</sup>, hTSPO<sup>A147T</sup> and non-transfected (WT) mitochondria that were immunoprecipitated by either V5 antibodies or control IgG.** 647 unique proteins were identified, based on 1 or more unique peptide(s) with total spectrum count >2 and with a protein False Discovery Rate ≤1%.
(XLSX)

## Author Contributions

**Conceptualization:** Yazi D. Ke, Lars M. Ittner.

**Data curation:** Prita R. Asih, Anne Poljak.

**Formal analysis:** Prita R. Asih, Anne Poljak.

**Funding acquisition:** Michael Kassiou, Yazi D. Ke, Lars M. Ittner.

**Methodology:** Prita R. Asih, Anne Poljak.

**Resources:** Michael Kassiou.

**Supervision:** Michael Kassiou, Yazi D. Ke, Lars M. Ittner.

**Visualization:** Prita R. Asih.

**Writing – original draft:** Prita R. Asih, Lars M. Ittner.

**Writing – review & editing:** Prita R. Asih, Anne Poljak, Michael Kassiou, Yazi D. Ke, Lars M. Ittner.

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
