## [Decision Letter · Decision Letter 0]

27 Jul 2021

PONE-D-21-18913

Differential mitochondrial protein interaction profile between human translocator protein and its A147T polymorphism variant

PLOS ONE

Dear Dr. Ittner,

Thank you for submitting your manuscript to PLOS ONE. After careful consideration, we feel that it has merit but does not fully meet PLOS ONE’s publication criteria as it currently stands. Therefore, we invite you to submit a revised version of the manuscript that addresses the points raised during the review process.

We look forward to receiving your revised manuscript.

Kind regards,

Anjani Kumar Tiwari, Ph.D.

Academic Editor

PLOS ONE

Journal Requirements:

This works was supported by funding from the National Health and Medical Research Council 

13 and the Australian Research Council to M.K., Y.D.K. and L.M.I..

Additional Editor Comments:

This is very good attempt to see the interactions, but results are not very promising. Further supporting data are required . Author may give insights of additional studies if possible and should add a detailed future studies required in this directions. It will help to those who are trying to establish new insights for this protein in different pathological conditions.

Reviewers' comments:

Reviewer's Responses to Questions

**Comments to the Author**

1. Is the manuscript technically sound, and do the data support the conclusions?

Reviewer #1: Yes

Reviewer #2: Yes

2. Has the statistical analysis been performed appropriately and rigorously? 

Reviewer #1: No

Reviewer #2: Yes

3. Have the authors made all data underlying the findings in their manuscript fully available?

Reviewer #1: Yes

Reviewer #2: No

4. Is the manuscript presented in an intelligible fashion and written in standard English?

Reviewer #1: Yes

Reviewer #2: Yes

5. Review Comments to the Author

Reviewer #1: The manuscript of the paper entitled "Differential mitochondrial protein interaction profile between human translocator protein and its A147T polymorphism variant' is presented in a proper manner. However, author should consider following corrections before resubmission:

1.Page 10, L16 replace ‘cOmplete with ‘complete’

2.Page 10, L17; Page 10, L 22; Page 11, L 6, replace ‘))’ with ‘)’

3.Page 11, L 13 replace ‘inhibitor .’ with ‘inhibitor.’

4.Maintain consistency in units such as 50 mM instead of 50mM, 4oC instead of 4o C, min instead of Minutes, h instead of hour, either mL or ml, g instead of gram, 10 cm instead of 10cm, μl instead of microlitres etc

5.Page 12, L 13-15, L16-18 correct the sentences

6.Page 17, L12-14, reframe the sentence

7.Page 20, line 8, Replace ‘experiments .’ with ’experiments.’

I found the experiments were executed in a planned way. Technically manuscript is good.

Reviewer #2: An interesting research work by Lars M Ittner and team.

This is first such study to see interaction profile between human translocator protein and its A147T polymorphism variant.

In summary, the results identified the loss of a subset of interaction partners with the 6 A147T polymorphism variant of human TSPO but one of the positive aspects seems

"TSPO’s interaction with YWHAQ 7 might contribute to the regulation of TSPO’s targeting into mitochondrial membranes involved 8 in apoptotic signaling pathways"

Does author further support it by additional evidence.

6. PLOS authors have the option to publish the peer review history of their article (what does this mean?). If published, this will include your full peer review and any attached files.

Reviewer #1: **Yes: **Dr Pooja Srivastava

Reviewer #2: No

---

## [Author Response · Author response to Decision Letter 0]

7 Feb 2022

Point-to-point response to reviewer comments

We thank both reviewers for their comments that have been addressed as indicated below. We hope you find the revised version of the manuscript ready for publication in PLOS ONE.

Reviewer #1: The manuscript of the paper entitled "Differential mitochondrial protein interaction profile between human translocator protein and its A147T polymorphism variant' is presented in a proper manner. However, author should consider following corrections before resubmission:

1.Page 10, L16 replace ‘cOmplete with ‘complete’

2.Page 10, L17; Page 10, L 22; Page 11, L 6, replace ‘))’ with ‘)’

3.Page 11, L 13 replace ‘inhibitor .’ with ‘inhibitor.’

4.Maintain consistency in units such as 50 mM instead of 50mM, 4oC instead of 4o C, min instead of Min, h instead of h, either mL or ml, g instead of gram, 10 cm instead of 10cm, μl instead of microlitres etc

5.Page 12, L 13-15, L16-18 correct the sentences

6.Page 17, L12-14, reframe the sentence

7.Page 20, line 8, Replace ‘experiments .’ with ’experiments.’

Response: We thank this reviewer for the detailed comments and have made the requested changes. All changes have been highlighted in the marked version of the revised manuscript for your convenience.

I found the experiments were executed in a planned way. Technically manuscript is good.

Response: This is very much appreciated.

Reviewer #2: An interesting research work by Lars M Ittner and team.

This is first such study to see interaction profile between human translocator protein and its A147T polymorphism variant.

In summary, the results identified the loss of a subset of interaction partners with the 6 A147T polymorphism variant of human TSPO but one of the positive aspects seems

"TSPO’s interaction with YWHAQ 7 might contribute to the regulation of TSPO’s targeting into mitochondrial membranes involved 8 in apoptotic signaling pathways"

Does author further support it by additional evidence.

Response: We have carefully reworded this sentence to indicate that future investigation is needed to provide direct evidence. It reads now: “Based on DAVID Functional Clustering enrichment analysis, a possible role of the TSPO’s interaction with YWHAQ could be the regulation of TSPO’s targeting into mitochondrial membranes involved in apoptotic signaling pathways. But future studies are required to provide direct evidence.”

---

## [Decision Letter · Decision Letter 1]

6 Apr 2022

Differential mitochondrial protein interaction profile between human translocator protein and its A147T polymorphism variant

PONE-D-21-18913R1

Dear Author

We’re pleased to inform you that your manuscript has been judged scientifically suitable for publication and will be formally accepted for publication once it meets all outstanding technical requirements.

Kind regards,

Anjani Kumar Tiwari, Ph.D.

Academic Editor

PLOS ONE

Additional Editor Comments (optional):

This form of manuscript is ok for publication.

Reviewers' comments:

Reviewer's Responses to Questions

**Comments to the Author**

Reviewer #2: All comments have been addressed

Reviewer #3: All comments have been addressed

2. Is the manuscript technically sound, and do the data support the conclusions?

Reviewer #2: Yes

Reviewer #3: Yes

3. Has the statistical analysis been performed appropriately and rigorously? 

Reviewer #2: N/A

Reviewer #3: Yes

4. Have the authors made all data underlying the findings in their manuscript fully available?

Reviewer #2: Yes

Reviewer #3: No

5. Is the manuscript presented in an intelligible fashion and written in standard English?

Reviewer #2: No

Reviewer #3: Yes

6. Review Comments to the Author

Reviewer #2: As i can see that nearly all the questions asked by experts have been replied in revised manuscript therefore in my opinion Academic Editor may permit for publication in PLOS ONE

Reviewer #3: All the queries raised by previous reviewers have been replied, therefore it can be considered for publication in PLOS One.

7. PLOS authors have the option to publish the peer review history of their article (what does this mean?). If published, this will include your full peer review and any attached files.

Reviewer #2: No

Reviewer #3: No

---

## [Editor Report · Acceptance letter]

28 Apr 2022

PONE-D-21-18913R1 

Differential mitochondrial protein interaction profile between human translocator protein and its *A147T* polymorphism variant 

Dear Dr. Ittner:

I'm pleased to inform you that your manuscript has been deemed suitable for publication in PLOS ONE. Congratulations! Your manuscript is now with our production department. 

Kind regards, 

on behalf of

Dr. Anjani Kumar Tiwari 

Academic Editor

PLOS ONE